# NMDA receptor activation induces long-term potentiation of glycine synapses

**Michelle L. Kloc**¤a☉, **Bruno Pradier**¤b☉, **Anda M. Chirila**¤c, **Julie A. Kauer**[iD]¤d*

Dept. of Pharmacology, Physiology and Biotechnology, Brown Institute for Brain Science, Brown University, Providence, RI, United States of America

☉ These authors contributed equally to this work.
¤a Current address: Epilepsy, Cognition & Development Group, Department of Neurological Sciences, University of Vermont Larner College of Medicine, Burlington, VT, United States of America
¤b Current address: Department of Anesthesiology, Intensive Care and Pain Medicine, University Hospital, Muenster, Germany
¤c Current address: Department of Neurobiology, Harvard Medical School, Boston, MA, United States of America
¤d Current address: Department of Psychiatry & Behavioral Sciences, Stanford University School of Medicine, Stanford, CA, United States of America
* jkauer@stanford.edu, jkauer@stanford.edu

**Data Availability Statement:** The data are freely available at Figshare at this DOI: 10.6084/m9.figshare.9596552.

**Funding:** This work was supported by NIH NINDS grant NS050570 to Julie A Kauer (JAK), https://

## Abstract

Of the fast ionotropic synapses, glycinergic synapses are the least well understood, but are vital for the maintenance of inhibitory signaling in the brain and spinal cord. Glycinergic signaling comprises half of the inhibitory signaling in the spinal cord, and glycinergic synapses are likely to regulate local nociceptive processing as well as the transmission to the brain of peripheral nociceptive information. Here we have investigated the rapid and prolonged potentiation of glycinergic synapses in the superficial dorsal horn of young male and female mice after brief activation of NMDA receptors (NMDARs). Glycinergic inhibitory postsynaptic currents (IPSCs) evoked with lamina II-III stimulation in identified GABAergic neurons in lamina II were potentiated by bath-applied $Zn^{2+}$ and were depressed by the prostaglandin $PGE_2$, consistent with the presence of both GlyRα1- and GlyRα3-containing receptors. NMDA application rapidly potentiated synaptic glycinergic currents. Whole-cell currents evoked by exogenous glycine were also readily potentiated by NMDA, indicating that the potentiation results from altered numbers or conductance of postsynaptic glycine receptors. Repetitive depolarization alone of the postsynaptic GABAergic neuron also potentiated glycinergic synapses, and intracellular EGTA prevented both NMDA-induced and depolarization-induced potentiation of glycinergic IPSCs. Optogenetic activation of trpv1 lineage afferents also triggered NMDAR-dependent potentiation of glycinergic synapses. Our results suggest that during peripheral injury or inflammation, nociceptor firing during injury is likely to potentiate glycinergic synapses on GABAergic neurons. This disinhibition mechanism may be engaged rapidly, altering dorsal horn circuitry to promote the transmission of nociceptive information to the brain.

www.ninds.nih.gov/. The funders had no role in study design, data collection and analysis, decision to publish, or preparation of the manuscript.

**Competing interests:** The authors have declared that no competing interests exist.

## Introduction

In the superficial dorsal horn, thermal, mechanical, and nociceptive information is processed and then conveyed to the brain via ascending inputs. The wiring diagram of nociceptive information flow in the dorsal horn is far from complete, but inhibitory synapses have long been recognized as important points of control restricting the transmission of pain information to the brain [1–7]. Pharmacological disinhibition allows peripheral afferents from low-threshold mechanosensory cells that usually activate lamina III-V neurons to drive lamina I projection neurons *in vitro* [2, 8]. Glycine receptors (GlyRs) are most prevalent in caudal brain regions and in the spinal cord, where they mediate a large proportion of inhibitory neurotransmission [9, 10]. Either acute blockade [6, 11] or chronic loss [12–14] of glycinergic transmission in the spinal cord results in allodynia, hyperalgesia, and itch, while a GlyR allosteric modulator reduces neuropathic pain [15].

Glycinergic neurons in the dorsal horn and brainstem trigeminal nucleus innervate both excitatory and inhibitory neurons in the superficial layers [16, 17]. GlyRs are ligand-gated ion channels, members of the cys-loop superfamily that includes $GABA_A$ receptors, $5HT_3$ receptors, and nicotinic nACh receptors. Like $GABA_A$ receptors, glycine receptors are chloride channels and generally act to hyperpolarize cells and stabilize the membrane potential. Glycine receptors exist as heteromeric pentamer complexes of alpha ($\alpha$1–4) and $\beta$ subunits. The $\beta$ subunits bind to the scaffolding protein, gephyrin to stabilize the receptors at synapses [18]. After very early postnatal development, synaptic GlyRs in the dorsal horn are heteromers composed of $\alpha$1, $\alpha$3, and $\beta$ subunits [19–21], and $\alpha$3 subunits in the dorsal horn may be preferentially required for nociception [15, 19].

Synaptic plasticity in the nociceptive circuitry has the potential to switch the circuit rapidly from a resting state to a pain state, and understanding mechanisms of plasticity is therefore of interest both in normal nociception and pain states and in pathological pain such as neuropathic pain. Long-term potentiation (LTP) is a characteristic of many excitatory brain synapses, and has also been reported at glutamatergic synapses made by primary afferents in the dorsal horn [22–24], and GABAergic synapses in the dorsal horn have been shown to undergo LTP as well [25]. However, nearly nothing is known about mechanisms of plasticity at glycinergic synapses. We reported previously that LTP is induced at glycinergic synapses on GABAergic neurons in the dorsal horn by the proinflammatory cytokine interleukin 1 β, (IL-1β) [26], which is released in the dorsal horn following injury [27–29]. The same glycinergic synapses were maximally potentiated shortly after *in vivo* inflammation, and we hypothesized that glycine receptor LTP in this model was caused by local release of IL-1β during peripheral inflammation. Here we have identified other mechanisms that potentiate glycinergic synapses.

In cultured spinal cord neurons, NMDA can increase glycinergic currents [30]. Single-particle tracking experiments showed that clusters of GlyRs and miniature IPSC amplitudes are markedly increased after treatment with NMDA, but receptor clustering is prevented if $Ca^{2+}$ is chelated [31]. Elevation of intracellular $Ca^{2+}$ was also reported to potently increase glycine receptor single channel openings [32] in cultured cells or when heterologously expressed. Because of the relative paucity of information about glycinergic synapse plasticity and its potential importance in modulating nociception, we are interested in characterizing glycinergic synapses and the control of their synaptic strength *in situ* in the dorsal horn. We find that bath-applied NMDA causes a long-lasting potentiation of these glycinergic synapses through a postsynaptic mechanism. Simply depolarizing GABAergic neurons repetitively also potentiates glycinergic synapses, and both depolarization- and NMDA-induced potentiation are prevented by chelation of postsynaptic $Ca^{2+}$. Furthermore, NMDAR activation by primary nociceptors also potentiated glycinergic synapses. Together, our findings suggest that glutamate

released from primary nociceptive afferents during peripheral damage could act at NMDARs on inhibitory dorsal horn neurons to promote persistent potentiation of glycinergic synapses. These rapid onset synaptic changes are likely to contribute to nociceptive processing during normal pain states.

## Materials and methods

### Animals

All experiments were conducted in strict adherence to the National Institutes of Health Guide for the Care and Use of Laboratory animals and as approved by the Brown Institutional Animal Care and Use Committee. Animals included Tg(Gad2-EGFP)DJ31Gsat/Mmucd mice (GENSAT project, Rockefeller University; http://www.gensat.org), backcrossed more than ten times on the Swiss Webster background prior to use in this study. Hemizygous GAD65-EGFP mice were mated to Swiss Webster mice in each generation and were used as hemizygotes. Trpv1-Cre and lox-STOP-lox-ChR2-EYFP mice were purchased from The Jackson Laboratory. For optogenetic experiments, trpv1-Cre$^{+/+}$ mice were mated with ChR2-EYFP+/+ mice to generate trpv1$^{+/-}$/ChR2-EYFP$^{+/-}$ offspring (referred to here as TRPV1/ChR2). Both male and female mice of all genotypes (p25-p40) were maintained on a 12h light/dark cycle and were provided food and water ad libitum. Data taken from both male and female mice were included in this study, and no significant sex differences were identified. Animals were deeply anesthetized with isoflurane and then injected with a terminal dose of ketamine (75 mg/kg) and dexmedetomidine (1 mg/kg). Mice were then transcardially perfused with cutting solution containing (in mM): 92 choline chloride, 1.2 NaH$_2$PO$_4$, 1.2 NaHCO$_3$, 20 HEPES, 25 dextrose, 5 Na-ascorbate, 2 thiourea, 3 Na-pyruvate, 10 MgSO$_4$, 0.5 CaCl$_2$ [33, 34] that was bubbled with 95% O$_2$ and 5% CO$_2$. Animals were then decapitated, and the spinal cord was rapidly dissected from the ventral aspect. Transverse lumbar spinal cord slices (300 μm thick) were prepared as described previously [26]. Slices were incubated at 34˚C for 1 hour prior to recording in oxygenated recording ACSF containing (in mM) 119 NaCl, 2.5 KCl, 2.5 CaCl$_2$, 1 NaH$_2$PO$_4$, 1.3 MgSO$_4$, 26 NaHCO$_3$, 1.3 Na-ascorbate, 25 dextrose) and then stored at room temperature (RT) until use. Maintaining slices at RT improved their viability over time.

### Electrophysiology and optogenetics

Slices were continuously perfused with oxygenated ACSF at room temperature at a rate of 1 ml/min. To limit heterogeneity, recordings were restricted to the lateral area of dorsal horn lamina II. GABAergic neurons were visually identified, and only recordings from neurons that expressed GFP are included in this study, with the exception of experiments from trpv1-ChR2: optogenetic experiments were made from unlabeled lamina II neurons in slices from these mice. Current-clamp recordings were made at the start of every experiment to observe action potential firing patterns in response to current steps of 50 pA delivered at resting membrane potential every 10 seconds. Neurons with resting membrane potentials less than -55 mV or with holding currents greater than 50 pA were not considered healthy and were eliminated from further study.

Cells were voltage-clamped at -70 mV, and glycinergic IPSCs were evoked at 0.1 Hz using a stainless steel stimulating electrode placed lateral to the recording site in lamina II, and isolated using bicuculline (30 μM) and 6,7-dintroquinaloxine-2,3-dione (DNQX, 10 μM) to block GABA$_A$R and AMPAR currents, respectively. Remaining synaptic currents could be entirely blocked by strychnine confirming that they are glycinergic (see also Chirila et al., 2014). IPSCs were recorded as inward currents using pipettes filled with KCl-based internal solution containing (in mM): 125 KCl, 2.8 NaCl, 10 HEPES, 2 MgCl$_2$, 4 Na-ATP, 0.3 Na-GTP, 10 Na-phosphocreatine, 0.6 EGTA. In some experiments EGTA was increased to 15 mM in the pipette

solution, as noted. In these experiments, neurons were held for at least 20 minutes after breaking into whole-cell mode before another manipulation. For occlusion experiments NMDA (50μM) was bath applied for 5 min. If glycine inputs potentiated by more than 20%, IL-1β (10ng/ml, 10 min) was added to the bath at least 20 min following NMDA washout when IPSCs reached a steady state. For optogenetic experiments with low-frequency stimulation (LFS) of nociceptive inputs in TRPV1/ChR2 animals, an optic fiber (230μm diameter) was placed at the dorsal slice edge to stimulate TRPV1/ChR2 primary afferents with light (Plexon LED, 465nM, 0.5 - 1ms, 1-9mW). Presence of TRPV1/ChR2 input was determined in each slice/cell before bath application of DNQX. LFS was carried out in presence of bicuculline and DNQX; we stimulated each cell with a light train of 2 Hz for 2 min while voltage-clamped at -40mV at a 2ms light pulse duration. All pharmacological agents, including NMDA, were bath-applied at known concentrations unless otherwise indicated. Bicuculline and $PGE_2$ were dissolved in DMSO; the final concentration of DMSO in experimental ACSF solution was 0.03% and 0.1%, respectively. All other pharmacological agents were dissolved in water.

### Statistical analysis

All data are presented as mean ± SEM of the percent change in IPSC amplitude. Potentiation was measured at 12–17 minutes after NMDA application for NMDA-induced Gly LTP, or 15 minutes after the start of drug application. For PPR analysis, 60 IPSCs before and at 10–20 min after the start of NMDA or after depolarization were averaged; cells were included in PPR analysis if they exhibited at least 20% LTP above baseline values. We evaluated the effect experimental manipulation using one-way ANOVA (Fig 1) or paired t-tests (Figs 2–7) of non-normalized raw data at indicated time points. All statistical analyses were performed using GraphPad Prism software. Statistical tests were considered to be significant at a 95% confidence interval, with p values reported in the Results section.

## Results

We recorded from eGFP-labeled GAD-65 neurons in lamina II of the dorsal horn. This population of cells exhibited three main firing patterns in response to direct current injection (Fig 1A and 1B). The majority of neurons fired 1–3 action potentials in 300 ms (initial firing type, 60/141), while other cells exhibited either tonic firing throughout the pulse (54/141) or a delayed or "gap" mode of firing that is characteristic when A-type $K^+$ currents are present (27/141)(Yasaka et al., 2010). Cells we included in the gap/delay class sometimes fired an initial spike followed by a gap before resuming. The action potential threshold was significantly lower for tonic firing cells than for initial or gap/delay cells (Fig 1C; initial vs tonic, p<0.0001, gap vs. tonic, p<0.0001, one-way ANOVA, Tukey's multiple comparisons test; n = 60 initial, n = 54 tonic, n = 27 gap/delay), while the input resistance, and the rise time and decay time of evoked Gly IPSCs did not significantly differ among the classes (Fig 1D–1F).

### GlyRs in lateral dorsal horn lamina II contain both α1 and α3 subunits

Both GlyRα1 and GlyRα3 subunits have been reported in lamina II, and GlyRα3 in particular has been implicated in inflammatory pain [19]. Previous work has shown that $Zn^{2+}$ is an allosteric modulator of glycine receptors, transiently potentiating glycinergic synapses containing GlyRα1 [35, 36], while prostaglandin $E_2$ ($PGE_2$) depresses glycinergic synapses containing GlyRα3 [19]. Using these tools, we tested which receptor subunits were present in glycinergic synapses on the GABAergic neurons of lamina II. Glycinergic inhibitory postsynaptic currents (Gly IPSCs) were evoked using a stimulating electrode placed laterally nearby in lamina II. Bath-application of $ZnCl_2$ (1 μM) increased glycinergic IPSC amplitudes (Fig 2A–2C; 7/8

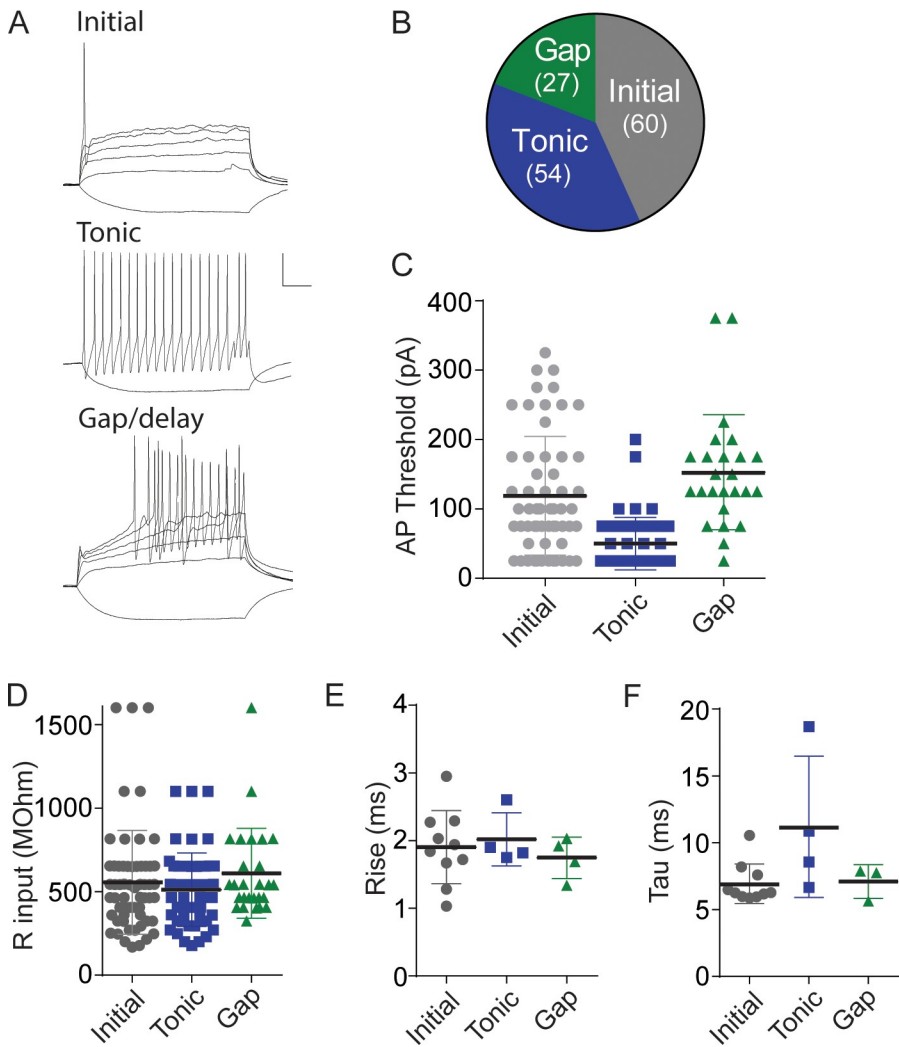

**Fig 1. Physiological characteristics of lamina II neurons labeled with eGFP in the eGFP-GAD-65 mouse line.** A. Three major firing patterns observed in eGFP-positive cells shortly after breaking in to whole-cell mode. Neurons were hyperpolarized and depolarized with 300 ms current steps of 50 pA. Top, initial firing (cells in this class also occasionally fired one or two more times during the 300 ms depolarization). Middle, tonic firing, showing only a single depolarizing response for clarity. Bottom, gap/delayed. Calibration: 20 mV, 50 ms. B. Chart illustrates the prevalence of these three firing types in our recordings. Gray, initial firing cells; blue, tonic firing cells; green, gap/delayed firing cells. Action potential threshold (C), input resistance (D), and glycine IPSC rise time (E) or decay time constant (F) are shown for cells in each class. Cells frequently received inputs with multiple apparent peaks, and these were necessarily excluded from rise/decay time analysis.

GABAergic neurons, 144±12.7% of baseline, p = 0.04, n = 8), while PGE$_2$ (10 µM) depressed glycinergic IPSCs in some but not all cells (Fig 2D–2F; 6/9 GABAergic neurons; 74±0.9% of baseline, p = 0.11, n = 9)(Fig 2). These data suggest that the majority of GABAergic neurons in our study are likely to have both GlyRα1- and α3-containing receptors at their synapses, while some may have only GlyRα1-containing receptors.

## NMDA potentiates Gly IPSCs

NMDAR activation increases glycinergic miniature IPSC amplitudes in cultures from embryonic spinal cord [30, 31]. To determine whether NMDAR activation similarly potentiates

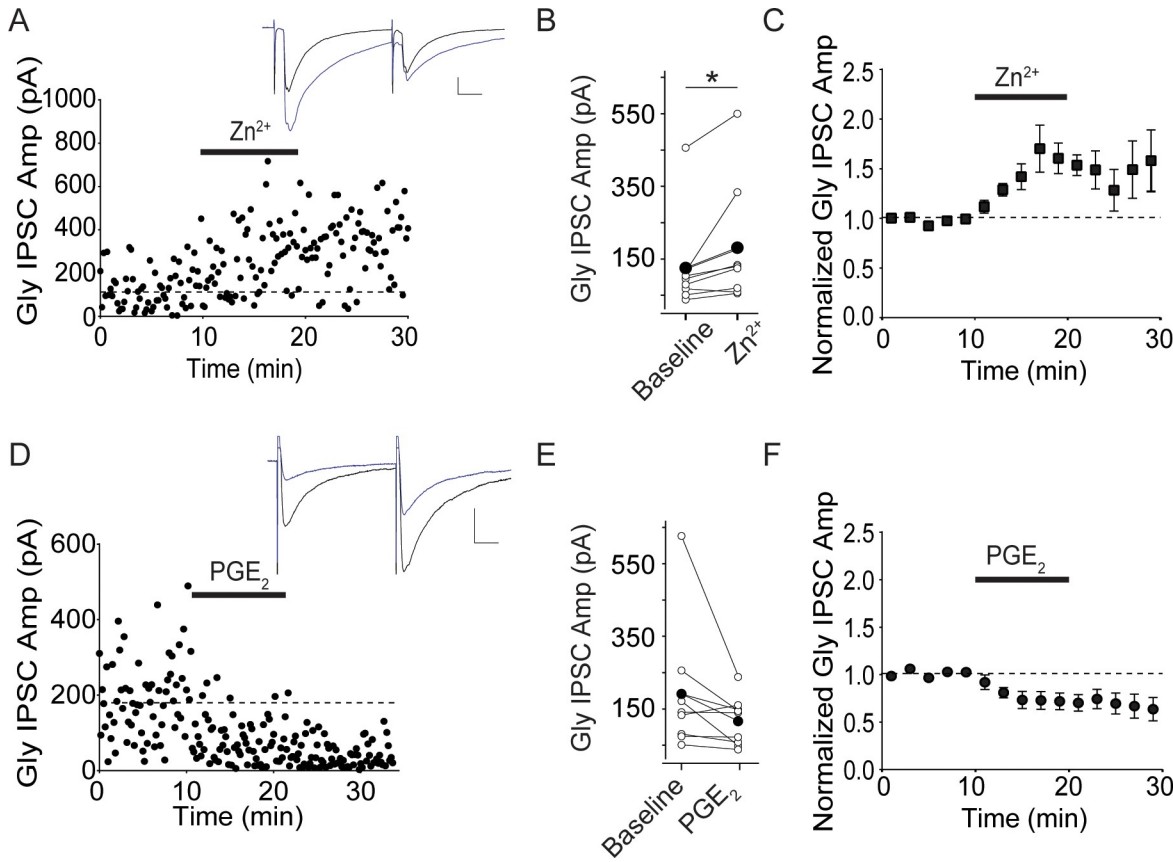

**Fig 2. Zn²⁺ potentiates Gly IPSCs and PGE₂ tends to inhibit Gly IPSCs in eGFP-GAD-65 lamina II neurons.** A. Example experiment illustrating Gly IPSCs recorded from an eGFP-positive lamina II neuron exposed to 1 μM ZnCl₂ for 10 minutes (bar). Inset: average of 5 IPSCs just before (black) and at 5 minutes after the end of ZnCl₂ application (blue). B. Raw data from all experiments of this type; bold bar and symbols represent the mean IPSC before and after ZnCl₂. C. Average of eight Zn²⁺ experiments. D. Example experiment illustrating Gly IPSCs recorded from an eGFP-positive lamina II neuron exposed to 10 μM PGE₂ for 10 minutes (bar). Inset: average of 5 IPSCs just before and at 5 minutes after the end of PGE₂ application. E. Raw data from all experiments of this type; bold bar and symbols represent the mean IPSC before and after PGE₂. F. Average of 9 PGE₂ experiments. Calibration: 100 pA, 10 ms.

glycinergic synapses *in situ* in the dorsal horn, we bath applied NMDA (50 μM, 5–10 minutes) and recorded evoked Gly IPSCs. NMDA application potentiated evoked Gly IPSC amplitudes within minutes (Fig 3A–3C; IPSC amplitudes: 147±7.6% vs. control amplitudes, p<0.0001, n = 40). Notably, NMDA potentiated Gly IPSCs in neurons of all three types of cell identified by action potential firing pattern (Fig 3D). Potentiation in most cells persisted throughout the recording period (up to 2 hours) after wash-out of NMDA; we therefore refer to this potentiation as NMDA-induced Gly LTP. The average decay time constant of Gly IPSCs was unchanged after NMDA, suggesting that the potentiation does not result from a decreased glycine transport (baseline $\tau$ = 6.98±0.43, post-NMDA $\tau$ = 7.46±0.45; p = 0.16, n = 7). To confirm that the potentiation was indeed produced via NMDA receptor activation, we bath-applied NMDA in the presence of the non-competitive NMDAR antagonist, 7-chlorokynurenic acid (100 μM) (Fig 3E–3G). As expected, NMDA did not potentiate glycine IPSCs under these conditions (IPSCs after 7-CK, 106±15.8% of baseline, p = .33, n = 4). In previous work, we reported that bath application of interleukin-1β (1L-1β) potentiates Gly IPSCs in lamina II gad2+ neurons (Chirila et al., 2014). We therefore tested whether NMDA-induced Gly LTP shares synaptic mechanisms with IL-1β-induced potentiation. We first bath-applied NMDA,

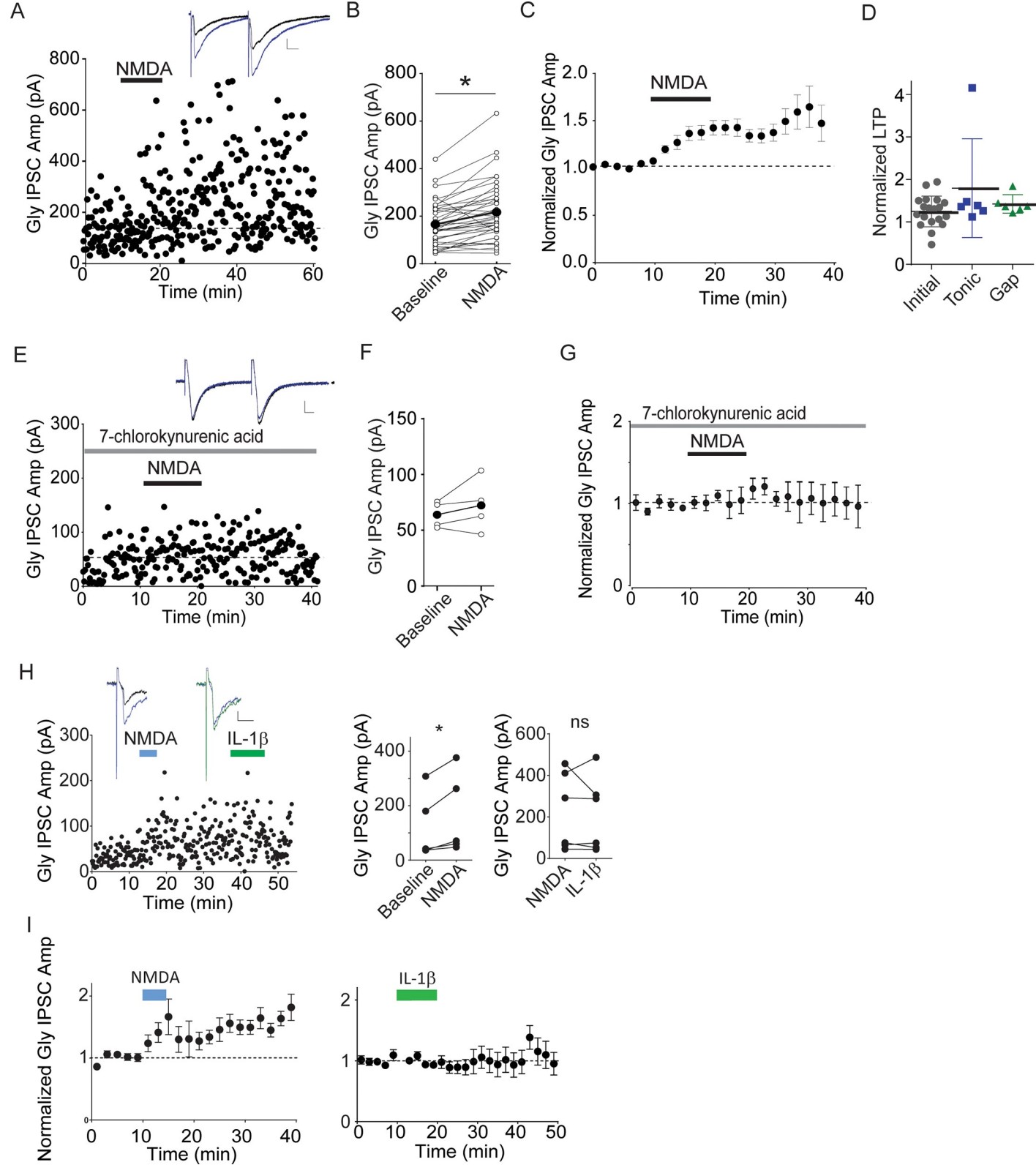

**Fig 3. NMDA receptor activation potentiates Gly IPSCs.** A. The mean IPSC before and after bath application of 50µM NMDA, 10 minutes. Inset: average of 5 IPSCs just before (black) and at 5 minutes after the start of NMDA application (blue). B. Raw data from all experiments of this type; bold bar and symbols represent the mean IPSC before and after NMDA. C. Average of 40 NMDA experiments. Fig 3B-C include experiments in which NMDA alone was bath-applied. Some of these

experiments were interleaved as control experiments for experimental treatments reported in other figures. D. The magnitude of NMDA Gly LTP measured at 10–15 minutes after the start of NMDA in the cell types described in Fig 1. E. Example experiment showing that Gly IPSCs pre-treated with 7-chlorokynurenic acid (gray bar) do not exhibit LTP after 50 μM NMDA (black bar). Inset: average of 5 IPSCs just before (black) and at 5 minutes after the start of NMDA application (blue). F. Raw data from each experiment of this type; bold bar and symbols represent the mean IPSC before and after NMDA. G. Averaged 7-Cl-kyn + NMDA experiments (n = 4). H. Application of NMDA occludes further potentiation by IL-1β. Left panel, single example. Inset: left, IPSC averages just before (black) and after NMDA (blue); right, after NMDA (blue) and after IL-1β (green). Right, single cell values before and 20 min after the start of NMDA; far right, 5 min after IL-1β. I. Averaged data from these experiments (n = 6). Calibration: 20pA, 10ms.

and then upon stable potentiation of Gly IPSCs, we applied IL-1β. As shown in Fig 3H–3J, after NMDA-induced potentiation (135±40%, n = 6, paired t-test baseline vs. NMDA: p = 0.03) IL-1β produced no further significant potentiation (93±32%, n = 6, paired t-test NMDA vs. IL-1β: p = 0.63), suggesting a shared underlying mechanism.

## NMDA-induced Gly LTP is mediated postsynaptically

LTP can result either from an increase in presynaptic neurotransmitter release or from an increase in postsynaptic receptor number or function, and the short-term dynamics of synaptic activation can be used to infer the locus of synaptic change. While measurements of miniature IPSCs can be used to suggest the locus, we found that mIPSCs occur so rarely in our recordings that they are not a very reliable method for collecting sufficient data. Instead, we measured the paired-pulse ratio (PPR), which generally decreases if the probability of neurotransmitter release increases during LTP [37–39], as well as postsynaptic responses to bath-applied glycine. On average, during Gly LTP the PPR remained unchanged after NMDAR application, suggesting that LTP is not caused by an increase in transmitter release, but instead by increased postsynaptic glycine receptor number or conductance (Fig 4A and 4B; PPR control, 1.18 ± 0.05; post-NMDA, 1.17 ± 0.07; p = .86, n = 26). To test this more directly, we measured whole-cell postsynaptic currents evoked by exogenous bath-applied glycine. These glycine currents are independent of presynaptic glycine release, thus any increase after NMDA can only result from postsynaptic changes. Glycine (3 mM) was applied for 30 s every ten minutes, and the resulting inward currents were recorded [26]. Control measurements were acquired by washing on glycine 2–3 times, and then NMDA (50 μM) was bath applied for five minutes. Following application of NMDA, exogenous glycine currents were significantly increased (Fig 4D and 4E; glycine currents after NMDA: 275 ± 30.5% of baseline, p = 0.011, n = 10). Together, our results suggest that NMDA Gly LTP depends upon an increase in number and/or function of GlyRs and is independent of glycine release.

## Postsynaptic calcium is required for NMDAR Gly LTP and sufficient to potentiate Gly IPSCs

Many forms of synaptic plasticity require a rise in postsynaptic calcium [26, 40, 41], and the high $Ca^{2+}$ permeability of the NMDAR is required for the majority of signaling through this channel. To determine whether NMDAR Gly LTP is calcium-dependent, we included EGTA (15 mM) in the recording pipette to chelate postsynaptic intracellular calcium. Compared with same-day control recordings, high intracellular EGTA blocked NMDAR Gly LTP (0.6 mM EGTA, 134 ± 12.06% of baseline, p = 0.03, n = 7; 15 mM EGTA, 95 ± 12.6% of baseline, p = 0.73, n = 8; Fig 5A–5C) suggesting that elevated intracellular calcium is necessary for NMDAR Gly LTP. To determine whether elevated $Ca^{2+}$ alone is sufficient to potentiate Gly synapses, GABAergic neurons were repetitively depolarized to -10 mV at 0.5 Hz for 10 minutes, to open VGCCs and elevate postsynaptic calcium [42, 43]. Synaptic stimulation was halted during this depolarization period. Following the depolarization protocol, Gly IPSCs were significantly increased (Fig 6A–6C; 150 ± 18.9% of baseline values, n = 11). Like NMDA-

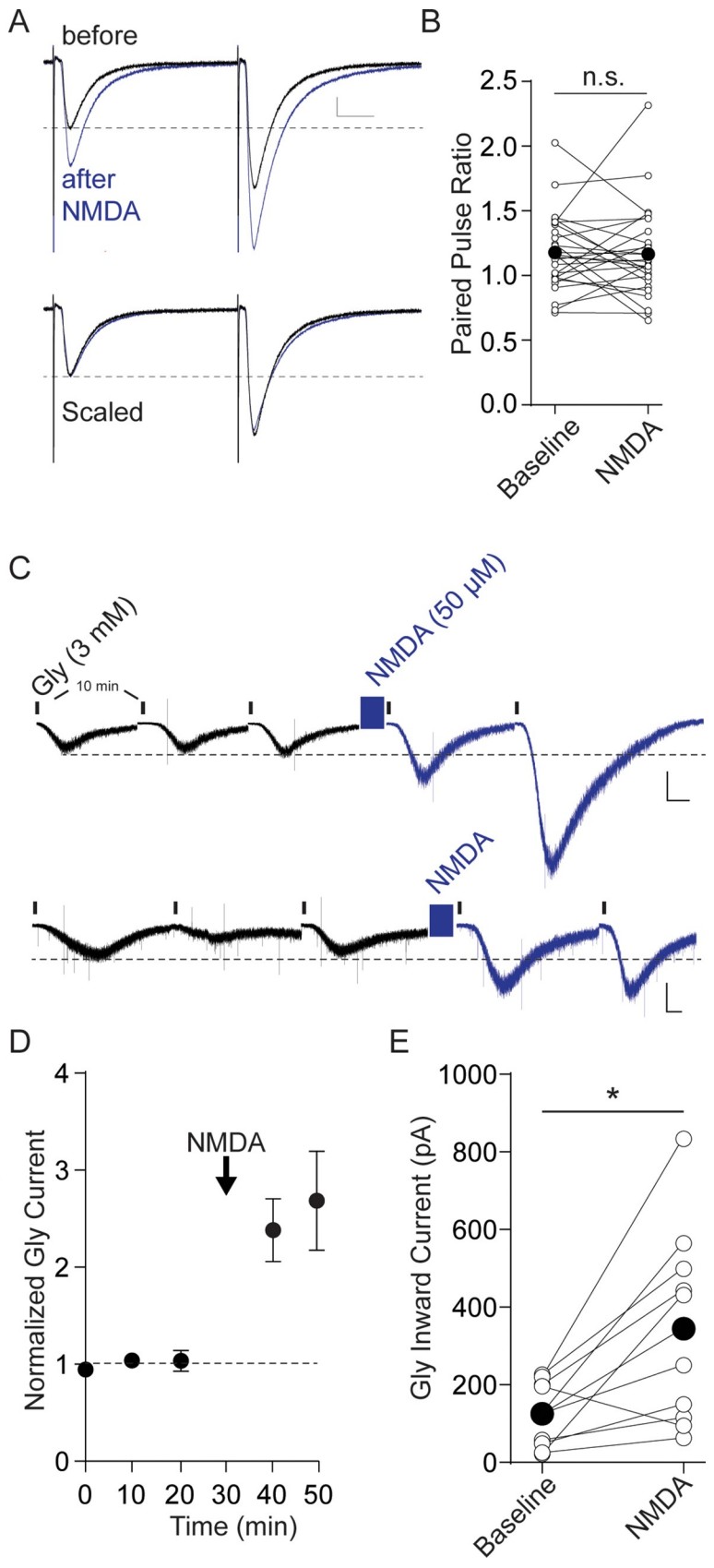

**Fig 4. NMDA does not alter PPR but does potentiate postsynaptic responses to glycine.** A. Paired pulse ratio does not decrease after NMDA-induced potentiation. Top, average of 5 IPSCs from a single experiment before (black) and 5 min after NMDA (blue). Bottom, the same IPSCs with the "after NMDA" trace scaled to the "before" trace to emphasize that PPR does not change. Calibration bar 50 pA, 10 ms. B. Raw data illustrating the PPR before and 10–20 min after NMDA application. Bold bar and symbols represent the mean PPR before and after NMDA. C. NMDA potentiates inward current responses to exogenously applied glycine. Recordings from two example experiments for which glycine (3 mM) was bath applied for 30 seconds once every 10 minutes (denoted by the small vertical bars). No synaptic stimulation was used in this experiment. NMDA (50 μM) application is marked by the blue box. Calibration bars: 250 pA, 20 s. Note the compressed time scale; between each illustrated response 10 minute segments are not shown for clarity. D. Averaged data from 10 such experiments normalized to 2–3 responses before NMDA. E. Raw data from each cell showing the magnitude of NMDA-induced potentiation. Bold bar and symbols represent the mean peak glycine current before and after NMDA.

induced Gly LTP, depolarization-induced potentiation was blocked by 15 mM intracellular EGTA (Fig 6B and 6C; 94 ± 10% of baseline values, n.s., n = 5; unpaired t-test depolarization alone vs. depolarization + 15 mM EGTA: p = 0.023). Together, the results suggest that postsynaptic $Ca^{2+}$ is necessary for NMDAR Gly LTP, and that postsynaptic calcium entry is also sufficient to potentiate glycinergic synapses.

## Brief low-frequency stimulation of nociceptor afferents potentiated Gly IPSCs through NMDA receptors

Sensitization of nociceptors is typically accompanied by an increased spontaneous discharge pattern of peripheral nociceptors, which induces changes in synaptic strength with dorsal horn neurons [23, 44]. While GlyR LTP was elicited by either bath-applied NMDA or experimenter-induced postsynaptic depolarization, both methods represent relatively unphysiological stimuli. To test our observations in a more physiological context, we next investigated whether Gly IPSCs could be potentiated upon activation of primary nociceptor afferents at frequencies occurring during injury or inflammation. We used TRPV1/ChR2 transgenic mice as recently described [44, 45] to optogenetically activate nociceptor afferents at a frequency that has been suggested to occur naturally during painful peripheral stimuli [23, 46]. Stimulation of nociceptors with light evoked glutamatergic synaptic events but not direct release of glycine [44]. We recorded Gly IPSCs in lamina II neurons in the presence of bicuculline and DNQX to allow isolated Gly IPSC recordings, however NMDARs were not blocked to permit activation of NMDARs by primary afferents. After a stable period of electrically-evoked Gly IPSCs, electrical stimulation was paused, and for 2 minutes 2 Hz light pulses were delivered (2ms, 9mW). Driving primary afferents in this manner potentiated Gly IPSCs evoked electrically once light stimulation had ended (Fig 7A, 129±17% of baseline values, n = 12, paired t-test baseline vs. after LFS: p = 0.002). The same experiment carried out in the presence of d-APV, however, did not trigger Gly LTP (Fig 7B, 98±26% of baseline values, n = 8, paired t-test baseline vs. LFS + APV: p = 0.87) and differed significantly from LFS-induced potentiation (unpaired t-test LFS vs. LFS + APV: p = 0.03).

## Discussion

Inhibitory synapses in the dorsal horn are prime potential sites of modulation during nociception. Our previous work demonstrated that glycine receptor synapses on GABAergic neurons in lamina II are potentiated 90 minutes after peripheral inflammation *in vivo*. Here we report that activation of NMDARs in spinal cord slices also potentiates glycinergic synapses on the same cells. The NMDA-induced potentiation is maintained by postsynaptic alterations in glycine receptors, as demonstrated by increased responsiveness to exogenously applied glycine.

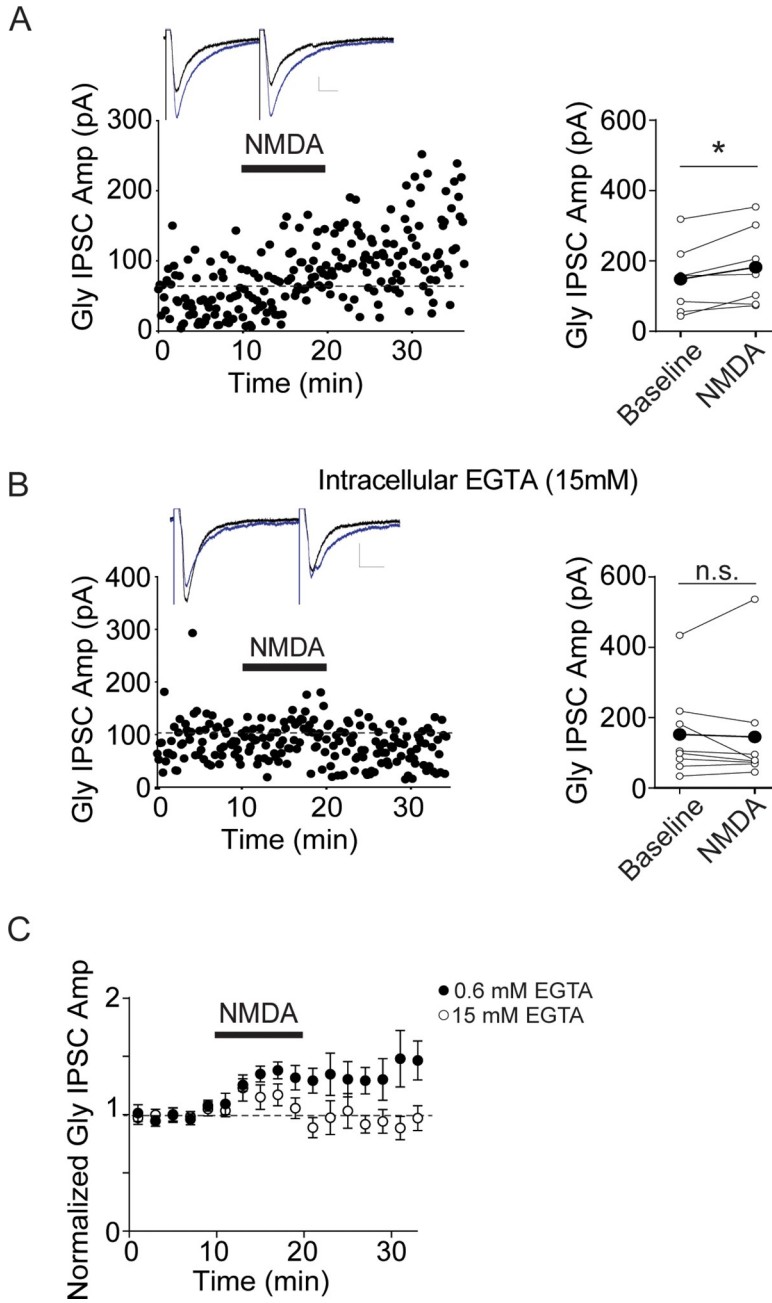

**Fig 5. NMDA Gly LTP is blocked by chelating postsynaptic Ca²⁺.** A. Left panel, example experiment showing Gly IPSC potentiation by NMDA in a control cell recording with 0.6 mM EGTA in the pipette solution. Inset, average of 5 IPSCs just before and 5 min after NMDA (blue). Right panel, raw data from 7 similar experiments. B. Left panel, example experiment showing Gly IPSCs before and after NMDA in a recording with 15 mM EGTA in the pipette solution. Inset, average of 5 IPSCs just before and 5 min after NMDA (blue). Right panel, raw data from 8 similar experiments. C. Averaged time course data from these experiments (0.6 mM EGTA, n = 7; 15 mM EGTA, n = 8). Experiments with each concentration of NMDA were carried out in alternation.

NMDA induced glycine potentiation requires a rise in intracellular Ca²⁺, and repetitive depolarization alone also potentiates the synapses in a Ca²⁺-dependent manner. Finally, NMDAR activation induced by synaptic stimulation of trpv1-lineage nociceptor afferents also potentiates glycinergic synapses.

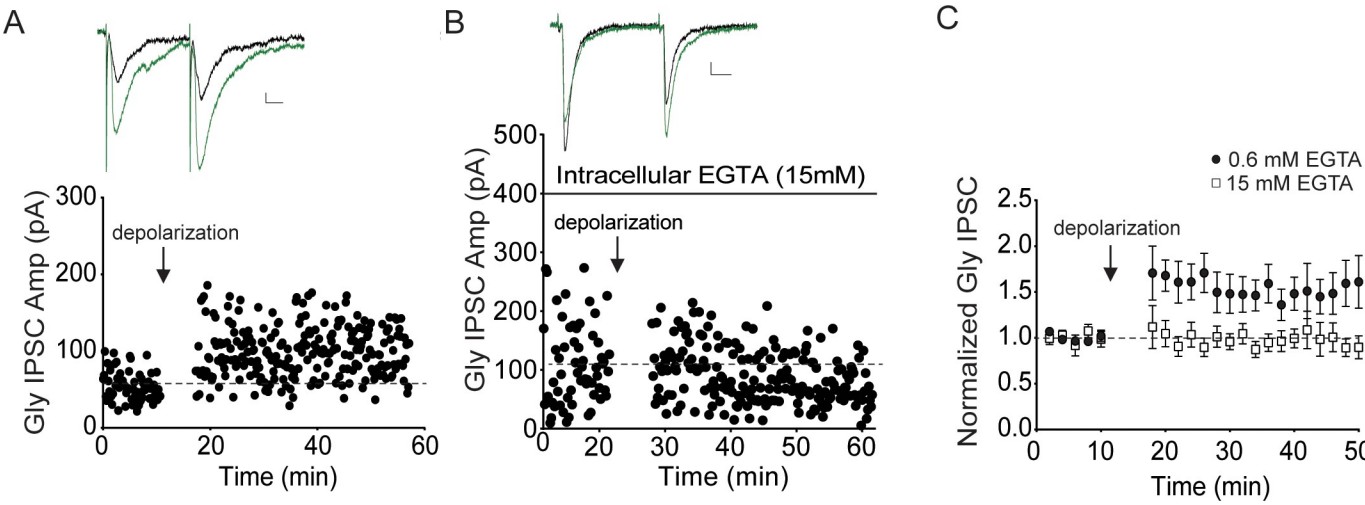

**Fig 6. Repetitive depolarization of the postsynaptic cell potentiates Gly IPSCs.** A. Example experiment showing that Gly IPSCs are potentiated after repetitive depolarizations (to -10 mV at 0.5 Hz for 10 minutes). Synaptic stimulation was halted during the depolarizations. Inset: average of 5 IPSCs just before and at 5 minutes after repetitive depolarization (green); calibration: 20 pA, 10 ms. B. Example experiment showing that inclusion of 15 mM EGTA in the recording pipette prevents depolarization-induced potentiation. C. Averaged time course of all experiments with either 0.6 mM (filled symbols, n = 12) or 15 mM EGTA (open symbols, n = 5) in the pipette solution.

## Firing properties of GAD-65 labelled lamina II neurons

Dividing dorsal horn neurons into functionally relevant subgroups has been crucial in beginning to define the circuitry of this complex and heterogeneous structure [47, 48]. Recent studies have used genetically labeled mouse lines to study a more homogeneous group and to begin to categorize distinct cell properties. In our experiments, we used neurons in lamina II labeled genetically with eGFP under the GAD-65 promoter that exhibited three distinct firing modes in response to depolarizing pulses (initial, tonic, and gap/delay). Previous work using either genetic labeling or peptide co-localization suggested that delayed or gap firing is a hallmark of excitatory interneurons, while tonic firing is more characteristic of inhibitory interneurons [49–52]. However, our data indicate that some neurons labeled in the GAD-65 reporter mouse exhibit delayed/gap firing, which has also been reported for a small subset of genetically identified inhibitory neurons in superficial layers of the spinal cord [51]. Here we have referred to these neurons as GABAergic for convenience, but recognize that approximately 20% of our recordings may represent another cell class since only 80% of neurons labeled in this mouse are GABA immunopositive [53]. Notably, in GAD-65 mice, only 60% of all lamina II GABAergic neurons are labelled, suggesting that studies using GAD-67 labeled neurons likely only sample a partially overlapping population [53]. GAD-65 labeled neurons (unlike most GAD-67 labeled cells) co-express c-fos after treatment with peripheral capsaicin, however, emphasizing the likely participation of the GAD-65 neurons we used in peripheral inflammatory processes [52].

## Glycine receptors on lamina II GABAergic neurons

Using $Zn^{2+}$ to probe for α1-containing receptors and $PGE_2$ to probe for α3-containing receptors, we found that all but one of our recorded glycinergic synaptic currents potentiated with $Zn^{2+}$, and the majority but not all exhibited synaptic depression with $PGE_2$. Our results suggest that glycinergic synapses on this GABAergic population have α1-containing receptors, but

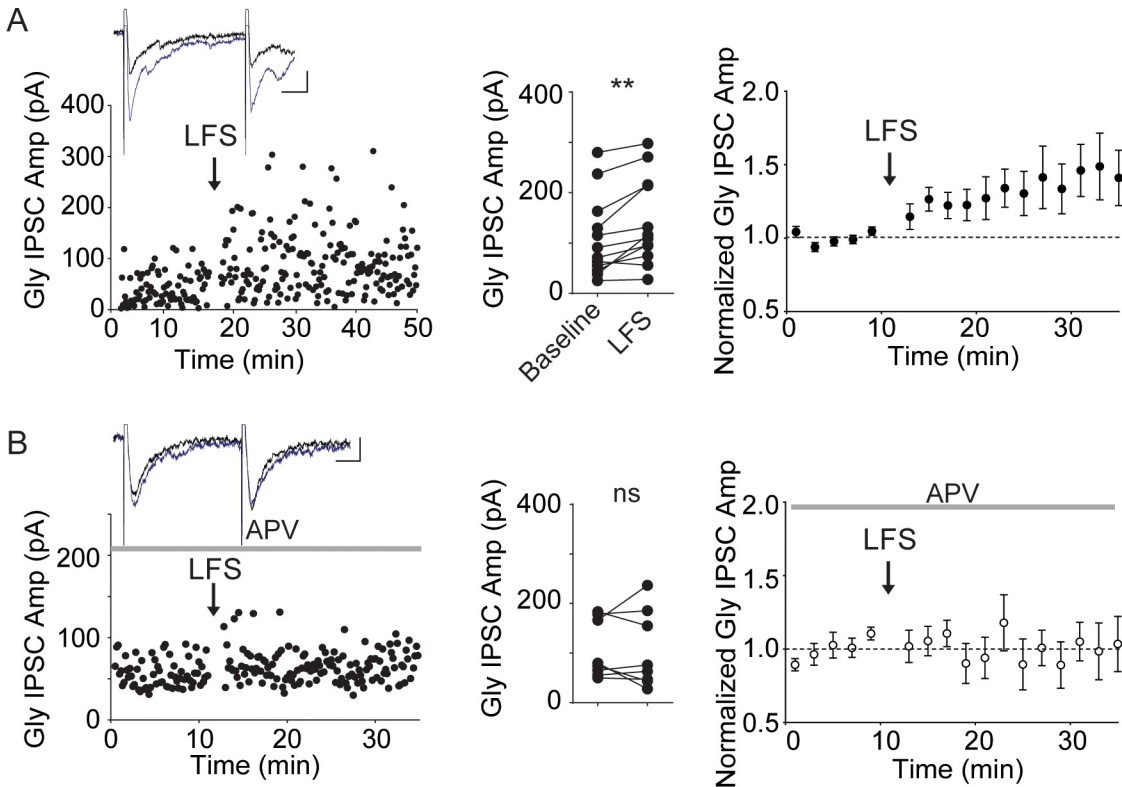

**Fig 7. Low frequency activation of primary nociceptor afferents potentiates glycinergic IPSCs in an NMDAR-dependent manner.** A. Left, example experiment illustrating electrical stimulation-evoked glycinergic IPSCs in a lamina II neuron before and after driving trpv1-lineage afferents using optical stimulation at 2 Hz for 2 minutes (LFS). Inset, averaged IPSCs taken just before (black) and 10 minutes after the start of LFS (blue). These glycinergic IPSCs were entirely blocked by 1 μm strychnine. Middle, individual data points for experiments of this type (n = 12). Right, average of 12 experiments. B. Similar experiments with 50 μM d-APV present throughout. Left, example experiment illustrating glycinergic IPSCs in a lamina II neuron before and after driving trpv1-lineage afferents using optical stimulation at 2 Hz. Middle, individual data points (n = 8). Right, average of 8 such experiments. Insets, calibration: 20pA, 10 ms.

may also contain α3-containing receptors, either as α1/α3/α heteromers or as α1α or α3α homomeric channels. Using immunocytochemistry, previous work indicated that approximately half of lamina II neurons appear to co-express both subunits, and our data in the GAD-65 cell population are consistent with these results [19]. Recent work in cultured neurons indicates that IL-1β does not affect GlyRα3-containing receptors, although IL-1β potentiation of GlyRα1-containing receptors was not observed in this study [54]. Our stimulation site in lamina II could activate glycinergic synapses from multiple sites, so it is also possible that distinct sets of afferents innervate synapses with differing GlyRα subtypes.

## NMDA-induced Gly LTP mechanisms

Brief bath-application of NMDA effectively potentiated glycinergic synapses on lamina II GABAergic neurons. The potentiation typically began within minutes of NMDA application, and persisted long after NMDA was washed out. We observed NMDA LTP in GAD-65-labeled neurons exhibiting a range of firing properties, including tonic firing and delayed firing. If single-spiking, tonic, and gap/delay cell populations in our study indeed represent functionally distinct groups, our results suggest that multiple postsynaptic cell types (including gap/delay cells) can exhibit postsynaptically-mediated glycinergic LTP. A non-competitive NMDAR

antagonist completely prevented LTP, indicating that LTP was not caused by off-target effects of NMDA. The potentiation was not accompanied by a significant decrease in the paired-pulse ratio, as expected if it were caused by an increased probability of glycine release. Instead, the LTP appears to result from a postsynaptic increase in glycine receptor number or function, as exogenously applied glycine currents were robustly increased after brief NMDA application. In this condition, the presynaptic release of glycine is not a factor, and instead this experiment confirms that NMDA treatment increases the postsynaptic response to glycine. The extracellular application of glycine might be expected to sample both synaptic and extrasynaptic glycine receptors. In considering the mechanism of potentiation, it is surprising to think of extrasynaptic glycine currents being enhanced after NMDA, as previous studies using single-particle tracking and other approaches have strongly suggested that glycine receptors are inserted and confined to synaptic regions where the scaffolding protein, gephyrin, is clustered [55, 56]. This observation of heterosynaptic plasticity is similar to the underlying mechanism of NMDAR-dependent homosynaptic LTP at AMPAR synapses, with receptors immobilized by synapse-to-cytoskeletal scaffolds [57]. The potentiation of bath-applied glycine responses we observe could therefore reflect insertion of glycine receptors at gephyrin-enriched synaptic sites. Alternatively, after NMDA application, extrasynaptic glycine receptors may also be inserted at sites expected to have low gephyrin levels. It is also possible that glycine receptors at all sites undergo an increase in single channel open times or affinity [32]; more work will be needed to distinguish these possibilities.

## Postsynaptic Ca$^{2+}$ and NMDA Gly LTP

Bath-application of NMDA has often been used to mimic neuronal activation and to induce NMDAR-dependent synaptic plasticity. For example, NMDA induces LTD or LTP at excitatory synapses [58–60] and can also potentiate GABAergic synapses heterosynaptically, via Ca$^{2+}$ [61–64] and calcium/calmodulin-dependent protein kinase II (CaMKII) [61, 64]. Similarly, glycinergic synapse strength is heterosynaptically potentiated by Ca$^{2+}$ influx and CaMKII [30, 55, 65, 66]. In cultured spinal cord neurons, GlyR clusters and miniature Gly IPSC amplitudes are both markedly increased after NMDA treatment; moreover, clustering was prevented when Ca$^{2+}$ was chelated [31]. Consistent with a similar mechanism, we found that high intracellular EGTA prevented NMDA-induced GlyR LTP. Repetitive depolarization of the postsynaptic cell alone also proved sufficient to potentiate glycinergic synapses, as long as intracellular Ca$^{2+}$ was not chelated; similar repetitive depolarization potentiates GABAergic synapses in visual cortex slices [67]. Importantly, low-frequency synaptic stimulation of primary nociceptor afferents was sufficient to potentiate glycinergic synapses on lamina II neurons, indicating that heterosynaptic NMDAR-dependent GlyR LTP is elicited with physiological stimuli. Together our data are consistent with the idea that a rise in intracellular Ca$^{2+}$ through NMDARs at nociceptor synapses, or even during action potential firing of the postsynaptic cell driven by any mechanism, can potentiate glycinergic synapses, most likely by augmenting synaptic glycine receptor numbers/function.

In our occlusion experiments testing whether IL-1β and NMDA potentiate glycinergic synapses via a similar mechanism, potentiation by IL-1β always prevented further potentiation by NMDA. We showed previously that potentiation by IL-1β is also prevented by EGTA (Chirila et al., 2014), and NMDA potentiation was also prevented by chelation of Ca$^{2+}$ by intracellular EGTA. These observations are consistent with a common final pathway for potentiation by both agents. Ninety minutes after peripheral inflammation of the paw, glycinergic synapses on GAD-65 labeled neurons are potentiated, via a postsynaptic mechanism [26]; we originally attributed this to local release of IL-1β after injury [27, 29]. Having shown here that NMDAR

activation either by NMDA or during primary afferent stimulation also potentiates these synapses, however, the inflammation-induced potentiation could occur through multiple mechanisms. Glutamate released from primary afferents at 2 Hz may elicit both release of IL-1β from dorsal horn glial cells and direct activation of NMDARs on lamina II neurons; our results show that both are expected to potentiate glycinergic synapses.

### Circuit considerations

What is the role of glycine receptor LTP in the synapses on lamina II GABAergic neurons in the nociceptive circuitry? We report that driving primary nociceptors for a brief period markedly potentiates glycine currents on lamina II neurons. Glycine receptor LTP in these neurons after peripheral inflammation [26] may serve to inhibit nociceptive information flow after injury, consistent with the fact that intrathecal strychnine promotes nocifensive behaviors [6, 68]. However, this method of applying strychnine increases the excitability of nearly all dorsal horn neurons; during NMDAR activation or inflammation, GlyR LTP occurring in GABAergic neurons may instead allow increased transmission of ascending nociceptive signals. The GABAergic interneurons of lamina II are often invoked as a component of the gate controlling the passage of peripheral nociceptive information to the brain [11, 69]. Potentiation of glycinergic synapses on inhibitory neurons by $Ca^{2+}$ influx via NMDARs or neuronal firing (or IL-1β release after inflammation) is expected to open the gate, effectively disinhibiting ascending nociceptive information flow. Reduced inhibition in the dorsal horn is observed in several animal models of persistent pain, suggesting that simply altering inhibitory synaptic function in the dorsal horn can mimic persistent pain syndromes [70–73]. Disinhibition also causes ascending projections that normally respond only to noxious stimuli to be activated by excitatory inputs carrying non-nociceptive signals (allodynia) [74, 75]. The capability of GABAergic interneurons to undergo Gly LTP, as well as their precise synaptic wiring and regulatory control of ascending neurons will determine the functional role of glycine receptor LTP in the dorsal horn. Our current understanding of the local circuit suggests that GAD-65 interneurons of lamina II sampled in our study could include islet cells or dynorphin-containing cells of lamina II, and possibly some parvalbumin neurons of lamina III [47]. Inhibition of any of these cell types could promote excitability of projection neurons in the dorsal horn, either directly or indirectly. If the gap/delay cells that underwent Gly LTP are excitatory interneurons as suggested by others [49–52], this adds another layer of complexity to the circuit possibilities. It will be critical in future work to identify how widespread the phenomenon of glycine receptor LTP is, as well as its behavioral consequences.

### Acknowledgments

The authors would like to thank Kauer lab members for helpful suggestions. We also are grateful for technical assistance from Ms. Ayumi Tsuda.

### Author Contributions

**Conceptualization:** Michelle L. Kloc, Bruno Pradier, Anda M. Chirila, Julie A. Kauer.

**Data curation:** Michelle L. Kloc, Bruno Pradier, Anda M. Chirila.

**Formal analysis:** Michelle L. Kloc, Bruno Pradier, Anda M. Chirila, Julie A. Kauer.

**Funding acquisition:** Julie A. Kauer.

**Investigation:** Michelle L. Kloc, Bruno Pradier, Anda M. Chirila.

**Supervision:** Julie A. Kauer.

**Writing – original draft:** Michelle L. Kloc.

**Writing – review & editing:** Bruno Pradier, Anda M. Chirila, Julie A. Kauer.

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
