## [Decision Letter · Decision Letter 0]

6 Aug 2019

PONE-D-19-19003

NMDA receptor activation induces long-term potentiation of glycine synapses

PLOS ONE

Dear Dr. Kauer,

Thank you for submitting your manuscript to PLOS ONE. After careful consideration, we feel that it has merit but does not fully meet PLOS ONE’s publication criteria as it currently stands. Therefore, we invite you to submit a revised version of the manuscript that addresses the points raised during the review process.

We would appreciate receiving your revised manuscript by Sep 20 2019 11:59PM. To enhance the reproducibility of your results, we recommend that if applicable you deposit your laboratory protocols in protocols.io, where a protocol can be assigned its own identifier (DOI) such that it can be cited independently in the future. For instructions see: http://journals.plos.org/plosone/s/submission-guidelines#loc-laboratory-protocols

We look forward to receiving your revised manuscript.

Kind regards,

Guangyu Wu, PhD

Academic Editor

PLOS ONE

Journal Requirements:

1. We note that you have indicated that data from this study are available upon request. PLOS only allows data to be available upon request if there are legal or ethical restrictions on sharing data publicly. For more information on unacceptable data access restrictions, please see http://journals.plos.org/plosone/s/data-availability#loc-unacceptable-data-access-restrictions.

Reviewers' comments:

Reviewer's Responses to Questions

**Comments to the Author**

1. Is the manuscript technically sound, and do the data support the conclusions?

Reviewer #1: Yes

Reviewer #2: Yes

2. Has the statistical analysis been performed appropriately and rigorously? 

Reviewer #1: Yes

Reviewer #2: I Don't Know

3. Have the authors made all data underlying the findings in their manuscript fully available?

Reviewer #1: Yes

Reviewer #2: Yes

4. Is the manuscript presented in an intelligible fashion and written in standard English?

Reviewer #1: Yes

Reviewer #2: Yes

5. Review Comments to the Author

Reviewer #1: In this study, the authors show that postsynaptic NMDA receptor activation, either via synaptic release of glutamate or exogenous application of NMDA, triggers long lasting potentiation of glycine receptor mediated synaptic events in GABAergic neurons in the dorsal horn. The experiments are quite conclusive and the conclusions are supported by the data presented. I see only a few minor issues that need clarification prior to publication.

1. The authors may have addressed this point in their earlier publications, but I think it is important to demonstrate that the putative glycinergic synaptic responses can be blocked by a classical antagonist such as strychnine

2. Is there an augmentation of glycinergic mIPSCs after NMDA treatment? Do their amplitudes increase as in evoked responses?

Reviewer #2: Recommendation: Minor revision

Comments to Author: Ref. PONE-D-19-19003

Title: NMDA receptor activation induces long-term potentiation of glycine synapses

Overview and general recommendation:

This manuscript mechanistically revealed how glycine synapses is potentiated by activation of NMDAR. I found that the paper is clearly written in general and much of it is well described. I feel confident that the authors performed careful and thorough experiments to test their hypothesis. However, the current version needs to be revised for publication due to the following reasons:

1. Missing certain literature citations in discussion part.

2. Please rewrite and emphasis the significance of this study in the manuscript.

6. PLOS authors have the option to publish the peer review history of their article (what does this mean?). If published, this will include your full peer review and any attached files.

Reviewer #1: No

Reviewer #2: No

---

## [Author Response · Author response to Decision Letter 0]

19 Aug 2019

Responses to the Review Comments to the Author

Again - please note that figures can only be seen in the letter we uploaded.

We are grateful for the reviewers’ comments, and have addressed each point below:

Reviewer #1: In this study, the authors show that postsynaptic NMDA receptor activation, either via synaptic release of glutamate or exogenous application of NMDA, triggers long lasting potentiation of glycine receptor mediated synaptic events in GABAergic neurons in the dorsal horn. The experiments are quite conclusive and the conclusions are supported by the data presented. I see only a few minor issues that need clarification prior to publication.

1. The authors may have addressed this point in their earlier publications, but I think it is important to demonstrate that the putative glycinergic synaptic responses can be blocked by a classical antagonist such as strychnine.

Thank you for this comment. We have now added in the methods that our responses are entirely blocked by 1 �M strychnine, and added the reference to a previous PNAS paper in which we showed this. I have attached the figure below for the convenience of the reviewer (Supplementary Figure 3b, Chirila et al., 2014). 

In addition, in one experiment in our more recent work, strychnine was added after NMDA and the synaptic current was entirely blocked (left-hand figure below). We also added strychnine at the end of 4 experiments in which glycinergic IPSCs were evoked in the trpv1-ChR2 animals – part of Figure 7 in the paper). As can be seen in the graphs (right hand figure above), IPSCs were strongly depressed by strychnine within ten minutes to less than 10% of basal values. We have now added this information to the manuscript, and also cited our previous experiment referring to the Chirila et al. paper.

2. Is there an augmentation of glycinergic mIPSCs after NMDA treatment? Do their amplitudes increase as in evoked responses?

We very much appreciate the reviewer’s comment. In the PNAS paper (Chirila et al., 2014) we very much wanted to examine miniature IPSCs; however, under our recording conditions, the minis are so rare that it was very difficult to collect a meaningful number within a reasonable time frame. In that paper, we therefore had to resort to using strontium solutions and a train of electrical stimuli to evoke asynchronous release, in order to produce a sufficient number of events before and after LTP to compare them. In the current manuscript, we felt that this approach would not add much to the paper. To test whether the LTP is mediated by pre- or postsynaptic changes, we measured both paired-pulse ratios (PPR; a sensitive measure of presynaptic probability of release) and exogenous delivery of glycine (a robust measure of postsynaptic sensitivity to glycine). As can be seen in Figure 4, there is zero change on average in PPR, supporting the idea that presynaptic release probability is unchanged, and moreover that the response to bath-applied glycine increases 2-3 fold over the time frame of our experiments, demonstrating that postsynaptic sensitivity is increased enormously after NMDA, consistent with our observations of LTP of glycinergic IPSCs. Given these strong data, to attempt difficult recordings of strontium-evoked asynchronous mIPSCs will not add much to our story. The best we could hope for in my view is to ascertain whether the frequency of minis increased (I am relatively certain that the amplitudes will increase); however, with strontium mIPSCs, claims about frequency are not as convincing as those of amplitude. In summary, we feel that the very large potentiation with simple bath-application of glycine after NMDA LTP, and lack of change in PPR are a sufficiently compelling demonstration of a postsynaptic LTP mechanism, and hope the reviewer will agree.

I will also note that in Figure 3i, we report that after NMDA application, IL-1� produces no further potentiation, supporting a common underlying mechanism. We know that IL-1� can increase strontium mIPSCs, and so this figure suggests that NMDA is likely to do so as well.

Reviewer #2: Recommendation: Minor revision

Comments to Author: Ref. PONE-D-19-19003

Title: NMDA receptor activation induces long-term potentiation of glycine synapses

Overview and general recommendation:

This manuscript mechanistically revealed how glycine synapses is potentiated by activation of NMDAR. I found that the paper is clearly written in general and much of it is well described. I feel confident that the authors performed careful and thorough experiments to test their hypothesis. However, the current version needs to be revised for publication due to the following reasons:

1. Missing certain literature citations in discussion part.

We are grateful to the reviewer for noting this error and have gone carefully over the entire paper with this point in mind. We found a reference that was not included in the discussion (Melzack and Wall, 1965), and have now added this citation. If the reviewer has other specific suggestions for citation issues, we are happy to make revisions.

2. Please rewrite and emphasis the significance of this study in the manuscript.

We have made changes in the introduction and discussion to address this important comment.

---

## [Editor Report · Decision Letter 1]

22 Aug 2019

NMDA receptor activation induces long-term potentiation of glycine synapses

PONE-D-19-19003R1

Dear Dr. Kauer,

We are pleased to inform you that your manuscript has been judged scientifically suitable for publication and will be formally accepted for publication once it complies with all outstanding technical requirements.

With kind regards,

Guangyu Wu, PhD

Academic Editor

PLOS ONE
---

## [Editor Report · Acceptance letter]

27 Aug 2019

PONE-D-19-19003R1 

NMDA receptor activation induces long-term potentiation of glycine synapses 

Dear Dr. Kauer:

I am pleased to inform you that your manuscript has been deemed suitable for publication in PLOS ONE. Congratulations! Your manuscript is now with our production department. 

With kind regards,

on behalf of

Dr. Guangyu Wu 

Academic Editor

PLOS ONE